# Enhanced production of l-fuculose by *Escherichia coli* engineered via genome-scale metabolic modeling

Gun-Hwi Yeon[1,2]☯, Du-Kyeong Kang[1,2]☯, Hyun-Jin Koo[3,4], Daewon Go[3,4], Jungyeon Kim[3,4]*, Bong Hyun Sung[1,2,5]*

1 Synthetic Biology Research Center, Korea Research Institute of Bioscience and Biotechnology (KRIBB), Daejeon, Republic of Korea, 2 Department of Biosystems and Bioengineering, Korea National University of Science and Technology (UST), Daejeon, Republic of Korea, 3 Graduate School of International Agricultural Technology, Seoul National University, Pyeongchang-gun, Gangwon-do, Republic of Korea, 4 Institute of Food Industrialization, Institutes of Green Bioscience and Technology, Seoul National University, Gangwon-do, Republic of Korea, 5 Graduate School of Engineering Biology, Korean Advanced Institute of Science and Technology (KAIST), Daejeon, Republic of Korea

☯ These authors contributed equally to this work.
* bhsung@kribb.re.kr (BHS); kim131812@snu.ac.kr (JK)

**Editor:** Jian Xu, Kyushu University Faculty of Agriculture Graduate School of Bioresource and Bioenvironmental Sciences: Kyushu Daigaku Nogakubu Daigakuin Seibutsu Shigen Kankyo Kagakufu, JAPAN

## Abstract

L-Fuculose is a rare deoxyketohexose sugar and is a structural isomer of L-fucose, which exhibits skin-lightening, moisturizing, and anti-aging effects. Due to their structural similarity, L-fuculose is also expected to provide potential health benefits. However, L-fuculose exists only in trace amounts in nature, making extraction from natural sources virtually impossible; to date, it has been synthesized mainly by enzymatic conversion. This approach, however, suffers from major limitations: both the substrate (L-fucose) and the enzyme involved (L-fucose isomerase) are costly; the enzymatic reaction cannot achieve complete conversion due to the chemical equilibrium; and the methods for purification of L-fuculose from the reaction mixture containing both L-fucose and L-fuculose are inefficient and uneconomical. Microbial cell factories have been explored as an alternative route for L-fuculose biosynthesis, but their production titers remain extremely low, limiting their industrial applicability. In this study, a microbial cell factory was engineered in *Escherichia coli* by redirecting the pathway of L-fucose metabolism toward L-fuculose production. Overexpression of *fucA* enabled the aldol condensation of lactaldehyde and dihydroxyacetone phosphate to produce L-fuculose-1-phosphate, which was subsequently dephosphorylated to L-fuculose by a sugar phosphatase. To prevent diversion of substrates and products into competing pathways, the *fucI*, *fucK*, *tpiA*, *fucO*, and *aldA* genes were deleted. The final engineered strain produced 50.25±4.30 mg/L of L-fuculose, a 32.4-fold increase compared to that achieved previously by microbial biosynthesis. This study establishes a foundation for the industrial production of L-fuculose, which has potential application as a valuable ingredient in cosmetics, functional foods, and pharmaceuticals.

**Data availability statement:** All relevant data are within the paper and its Supporting Information files.

**Funding:** This work was supported by National Research Foundation of Korea (NRF) grants funded by the Korean government (MSIT) (RS-2025-23282972, NRF-2022M3J5A1056169, 2021M3A9I5023254, 2019R1A2C1090726, and 2018M3A9H3024746); a National Research Council of Science & Technology grant (No. CAP20024-200) funded by the Korean government (MSIT); the Research Initiative Program of KRIBB; and the Korea Institute of Planning and Evaluation for Technology in Food, Agriculture, and Forestry (IPET) through the Technology Commercialization Support Program funded by the Ministry of Agriculture, Food, and Rural Affairs (MAFRA) (RS-2024-00401586). The funders had no role in study design, data collection and analysis, decision to publish, or preparation of the manuscript.

**Competing interests:** The authors have declared that no competing interests exist.

## Introduction

L-Fuculose, a rare sugar belonging to the deoxyketohexose family, is a structural isomer of L-fucose, a constituent of mucins [1,2]. L-fucose exhibits various health-promoting properties, including skin-lightening, moisturizing, and anti-aging effects [3–5]. Given their close structural similarity, L-fuculose is expected to perform comparable molecular functions [3]. L-fuculose serves as an important intermediate in the L-fucose metabolic pathway of gut microbiota [6–8]. Beyond its role in microbial energy production, it contributes to physiological processes, such as mucin binding, colonization, flagella metabolism, and carbohydrate utilization [9]. For example, L-fuculose-1-phosphate activates the expression of genes involved in ribose metabolism, thereby supporting microbial growth and activity [10]. These observations indicate that L-fuculose has potential health benefits and may also act as a signaling molecule in regulating gut microbial physiology.

Despite these promising features, the study and application of L-fuculose have been hindered by challenges in its production. L-fuculose exists only in trace amounts in nature, making extraction from natural sources virtually impossible [3]. To date, most L-fuculose production strategies have relied on enzymatic conversion using L-fucose isomerase (FucI) [3,11,12]. However, this method has considerable drawbacks: the substrate (L-fucose) and isomerase enzyme are expensive; the reaction is constrained by chemical equilibrium, preventing complete conversion of L-fucose to L-fuculose (with equilibrium favoring ~62.8% L-fucose to 28.2% L-fuculose); and the separation and purification of L-fuculose from the reaction mixture are labor-intensive and uneconomical. These limitations make enzymatic synthesis of L-fuculose unsuitable for industrial-scale production.

An alternative approach is to redirect the L-fucose metabolic pathway for L-fuculose biosynthesis. In this pathway, lactaldehyde (LAD) and dihydroxyacetone phosphate (DHAP) can be condensed by L-fuculose-1-phosphate aldolase (FucA) to L-fuculose-1-phosphate, which can then be dephosphorylated by sugar phosphatases to yield L-fuculose. Yeast has been used as a host and glucose and LAD were used as substrates in combination with FucA and FucIto produce L-fucose and L-fuculose [1]. However, this strategy yielded only trace amounts of L-fuculose (1.55 mg/L), and the co-production of L-fucose (7.75 mg/L) further complicated downstream purification.

To overcome these limitations, in the current study, a microbial cell factory in was constructed in *Escherichia coli*, one of the most widely-used industrial microorganisms, to produce L-fuculose through a redirected pathway of L-fucose metabolism. Specifically, *fucA* was overexpressed to enhance L-fuculose-1-phosphate synthesis, which was dephosphorylated by innate sugar phosphatases to generate L-fuculose. Competing metabolic routes were eliminated by deleting *fucI*, *fucK*, *tpiA*, *fucO*, and *aldA*. This study established a novel metabolic engineering strategy for the efficient and cost-effective biosynthesis of L-fuculose, laying the foundation for its potential application in cosmetics, functional foods, and pharmaceuticals.

## Materials and methods

### Genome-scale metabolic model (GEM) analysis to simulate production of fuculose-1-phosphate using *E. coli*

For GEM analysis, the core *E. coli* metabolic model was used to simulate growth [13]. The following reactions were included in the model to enable the production of L-fuculose: L-fuculose transport, L-fuculose production, L-fuculose phosphatase, L-fuculose kinase, fuculose-1-phosphate aldolase, 1,2-PDO oxidoreductase, and 1,2-PDO transport. MATLAB (version R2019b) and Cobra Toolbox (version 2.37.3) were used for analysis [14].

### Construction of mutants

To selectively remove the target genes from the genome of the MG1655 strain, a single-step gene knockout procedure was conducted [15]. The primers used in this study are described in S1 Table. Annealing regions (20 bp) homologous to the kanamycin resistance cassette template plasmid, pKD4, were included in addition to 50 bp of a sequence homologous to regions flanking the target genes. The purified polymerase chain reaction (PCR) product obtained was electroporated into MG1655, which had previously been transformed with the lambda Red recombinase expression vector, pKD46, bearing a heat-labile origin. To facilitate recombination of the electroporated fragment, expression of lambda Red recombinase in MG1655 cells was induced by treatment with 50 mM arabinose at 30 °C. After the transformation of the PCR fragment into MG1655 harboring pKD46, the cells were recovered in Luria-Bertani (LB) broth at 37 °C. Kanamycin-resistant transformants were grown at 37 °C to induce the loss of the heat-labile pKD46 plasmid. Loss of the pKD46 plasmid was confirmed by loss of resistance to ampicillin. Final knockout isolates were screened for the absence of target genes and the presence of the kanamycin cassette using PCR. The kanamycin cassette was removed via flanked FRT recombination sites to obtain a markerless mutant using pCP20.

### Construction of pCDF-*fucA* plasmid

The *fucA* fragment was amplified using PCR, with primer pairs *fucA* F/*fucA* R. The amplified PCR fragments were designed to share 15-nt homologous bases at each end. The pCDF Duet-1 plasmid was linearized using *Nde*I/*Xho*I restriction enzymes. The PCR fragments were fused with the plasmid DNA using the In-Fusion HD Cloning Kit (Clontech, Mountain View, CA, USA) to generate pCDF–*fucA*. Specifically, the In-Fusion cloning reactions were completed with 200 ng PCR fragment, 150 ng linearized plasmid DNA, 2.0 µL 5× In-Fusion HD Enzyme Premix, and double-distilled water (ddH$_2$O) for a final volume of 10 µL. The reaction mixture was incubated at 50 °C in a water bath for 15 min and then placed on ice for the transformation of competent DH5α *E. coli* cells. The pCDF–*fucA* plasmid was screened using colony PCR with the primer pair, pCDF V-F/V-R, and then sequenced.

### Culture conditions

Engineered *E. coli* strains were cultivated in 50-mL baffled flasks containing 17 mL of LB medium supplemented with 10 g/L glucose, 5 mM LAD, and 50 µg/mL streptomycin. Cultures were incubated at 30 °C with shaking at 180 rpm. Gene expression was induced with 0.2 mM isopropyl β-D-1-thiogalactopyranoside.

### Analyses of extracellular metabolites

Extracellular metabolites, including glucose, LAD, 1,2-propanediol (1,2-PDO), and acetic acid, were quantified using an Agilent 1100 HPLC system (Agilent Technologies, Santa Clara, CA, USA) equipped with a refractive index detector and an Aminex HPX-87H organic acid column (Bio-Rad, Hercules, CA, USA). Chromatographic separation was performed at 65 °C with 0.01 N H$_2$SO$_4$ as the mobile phase at a constant flow rate of 0.6 mL/min. The retention times were 8.8, 12.9, 14.8, and 17.1 min for D-glucose, LAD, acetic acid, and 1,2-PDO, respectively. Quantification was carried out using calibration curves generated from authentic analytical standards.

## Identification and quantification of l-fuculose

To identify and quantify ʟ-fuculose, an authentic ʟ-fuculose standard and supernatants of engineered *E. coli* cultures were analyzed using gas chromatography–mass spectrometry (GC/MS) after derivatization steps, such as methoximation and silylation. For methoximation, 20 μL of methoxyamine hydrochloride solution (40 mg/mL in pyridine) was added to the dried metabolite samples, which were then incubated at 30 °C for 90 min. Subsequently, silylation was carried out by adding 50 μL of *N*-methyl-*N*-trimethylsilyl-trifluoroacetamide to the mixtures, which were then incubated at 37 °C for 30 min.

Metabolite analysis was performed on an Agilent 7890A gas chromatograph coupled to a 5975C quadrupole mass spectrometer (Agilent Technologies). A 1.0-μL aliquot of the derivatized sample was injected in splitless mode into an RTX-5Sil MS column (30 m × 0.25 mm i.d.; 0.25 μm film thickness; Restek, PA, USA) equipped with a 10-m guard column. The oven temperature program was: initial hold at 50 °C for 1 min, ramping to 330 °C at 20 °C/min, and final hold at 330 °C for 5 min. Mass spectra were acquired at 85–550 m/z with electron impact ionization at 70 eV.

## Results and discussion

### *In silico* (GEM) analysis of *E. coli* for l-fuculose production

GEM analysis was conducted to predict the optimal flux distribution of *E. coli* for ʟ-fuculose biosynthesis when LAD and glucose were used as substrates. Under aerobic conditions, the *in silico* model predicted that *E. coli* would simultaneously utilize LAD and glucose via FucA activity to generate ʟ-fuculose-1-phosphate, which would subsequently be dephosphorylated to ʟ-fuculose by innate sugar phosphatases (Fig 1A). The simulation further indicated that glucose concentrations equal to or greater than those of LAD were required to sustain ʟ-fuculose biosynthesis (S2 Table), and that increasing oxygen availability would enhance both biomass formation and ʟ-fuculose production (Fig 1B). In contrast, under anaerobic conditions, all LAD would be diverted to 1,2-PDO through FucO activity, resulting in no flux toward ʟ-fuculose production (S2-S3 Table). These results suggest that sufficient oxygen supply is essential for efficient flux toward ʟ-fuculose synthesis.

GEM analysis also identified several competing metabolic pathways that could interfere with ʟ-fuculose biosynthesis. Although LAD can be converted to ʟ-fuculose-1-phosphate by FucA, it can also be diverted to 1,2-PDO by FucO or to lactic acid by AldA. The other substrate, DHAP, can be converted to glyceraldehyde-3-phosphate (G3P) by TpiA and directed into glycolysis. Furthermore, the synthesized ʟ-fuculose itself can be isomerized to ʟ-fucose by FucI or rephosphorylated to ʟ-fuculose-1-phosphate by FucK. Although a previous study demonstrated that innate sugar phosphatases were capable of dephosphorylating ʟ-fuculose-1-phosphate to ʟ-fuculose, low phosphatase activity could lead to the reconversion of the former into LAD and DHAP through the reverse FucA reaction [1]. Collectively, these results highlight the potential metabolic sinks that limit the efficient channeling of LAD and DHAP toward ʟ-fuculose production.

### Strain engineering of *E. coli* and l-fuculose production

Based on the aforementioned GEM analysis, gene deletions were introduced to reduce competing pathways and retain ʟ-fuculose as the final product. First, the *fucI* gene, which encodes an enzyme that converts ʟ-fuculose into its isomer (ʟ-fucose) and thereby generates a mixture of both sugars, was deleted. Second, the *fucK* gene, which encodes an enzyme that otherwise diverts ʟ-fuculose into central metabolism by phosphorylating it to ʟ-fuculose-1-phosphate, was deleted to enable the retention of ʟ-fuculose as the final product. In addition, *fucA* was overexpressed under the control of the T7 promoter to catalyze the aldol condensation of LAD and DHAP to L-fuculose-1-phosphate. Overexpression of *fucA* was confirmed by SDS-PAGE analysis following IPTG induction (S1 Fig), showing a distinct band at the expected molecular weight. The resulting strain, designated Fuc 1 (Δ*fucI* Δ*fucK*, *fucA* overexpression), was cultivated in glucose–LAD medium for 96 h.

A

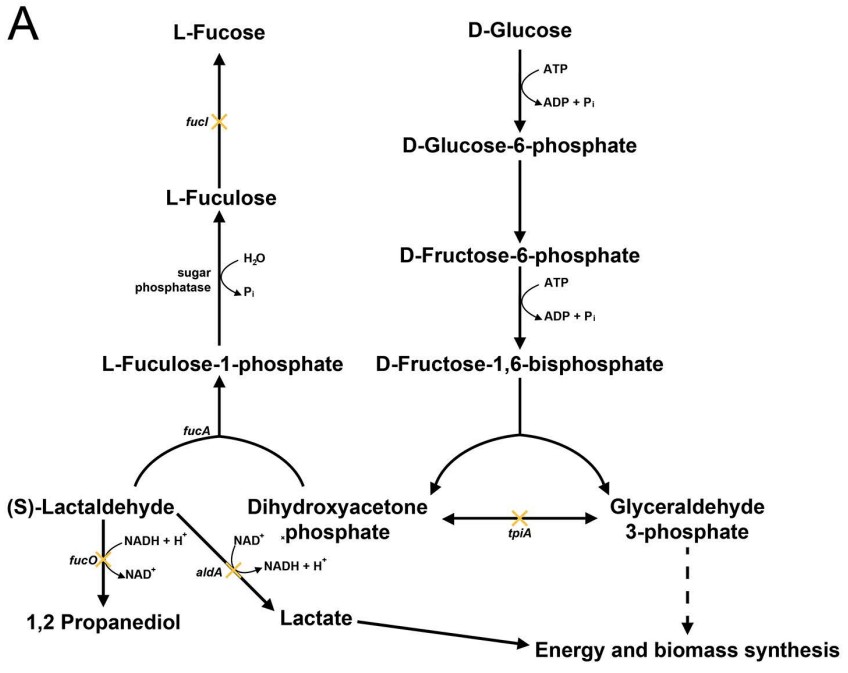

B

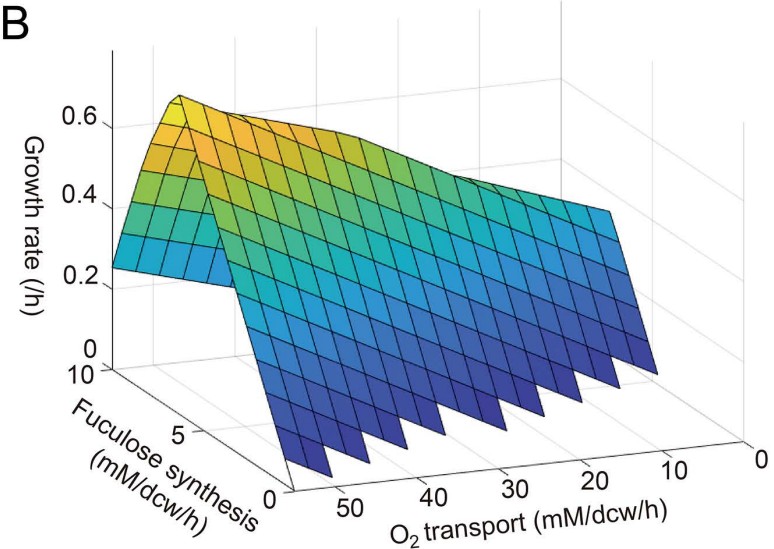

**Fig 1. *In silico* genome-scale metabolic model (GEM) of *Escherichia coli* for l-fuculose production. (A)** Simplified metabolic pathways of LAD and DHAP utilization leading to L-fuculose biosynthesis. Competing pathways and targeted gene deletions (*fucI, fucK, fucO, aldA, tpiA*) are indicated. **(B)** Predicted relationship between oxygen uptake rate, L-fuculose synthesis, and growth rate based on GEM analysis.

During fermentation, glucose was steadily consumed, cell density increased until 24 h and then entered the stationary phase, and acetic acid accumulated up to 24 h before stabilizing (Fig 2A). LAD was rapidly depleted within the first 12 h, whereas 1,2-PDO began to accumulate from 12 h, with its concentration increasing gradually until the end of fermentation. L-fuculose production was detected after 12 h and gradually increased throughout fermentation, reaching 38.92±5.19 mg/L (Fig 3A). GC/MS analysis of the culture supernatant confirmed the presence of L-fuculose (Fig 3B, C).

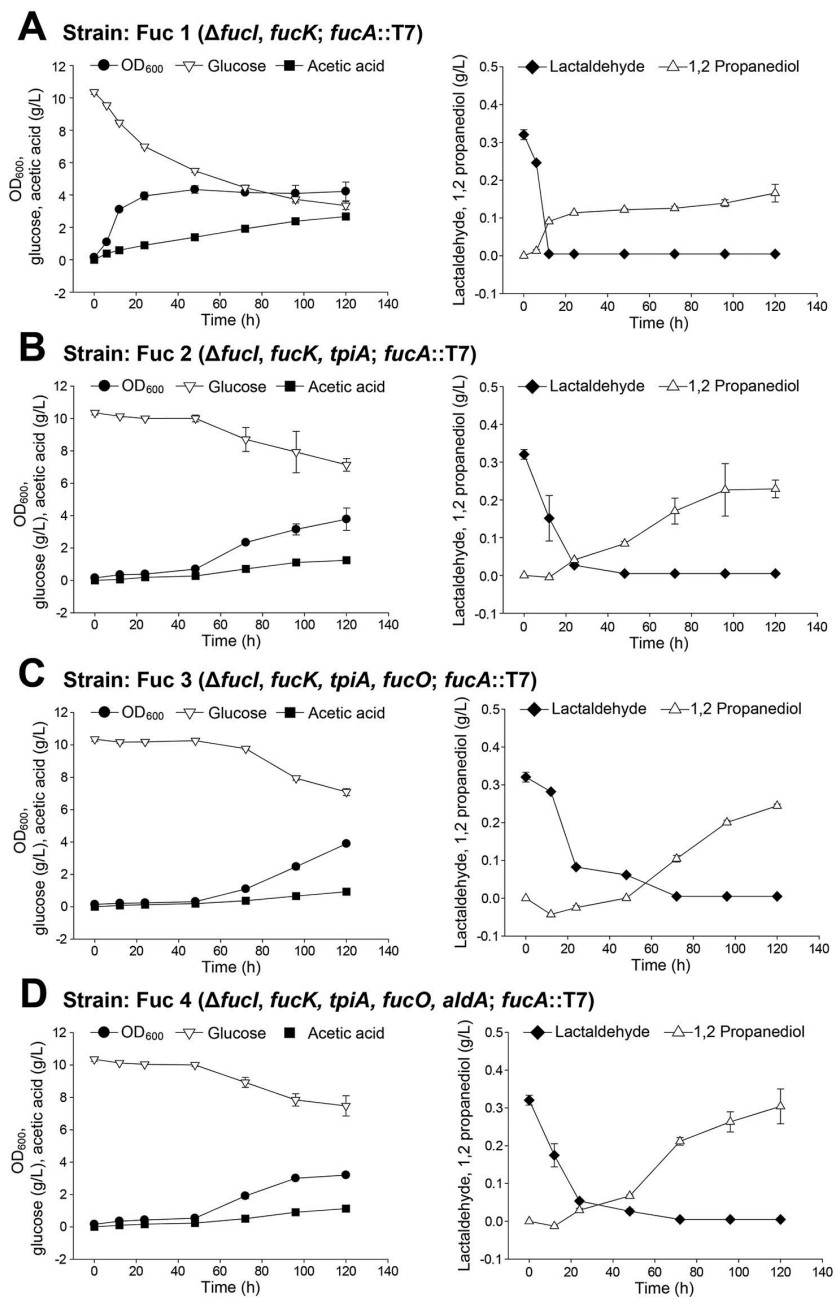

**Fig 2. Fermentation profiles of *E. coli* strains engineered for l-fuculose production. (A)** Fuc 1 (Δ*fucI* Δ*fucK*; *fucA*::T7). **(B)** Fuc 2 (Δ*fucI* Δ*fucK* Δ*tpiA*; *fucA*::T7). **(C)** Fuc 3 (Δ*fucI* Δ*fucK* Δ*tpiA* Δ*fucO*; *fucA*::T7). **(D)** Fuc 4 (Δ*fucI* Δ*fucK* Δ*tpiA* Δ*fucO* Δ*aldA*; *fucA*::T7). For each strain, cell growth (OD$_{600}$), glucose consumption, and acetic acid production are shown in the left panels, and LAD consumption with 1,2-propanediol (1,2-PDO) production is shown in the right panels. Error bars indicate standard deviations based on three independent experiments.

To increase intracellular availability of DHAP, the *tpiA* gene, which encodes an enzyme that converts DHAP to G3P for entry into glycolysis, was deleted in addition to *fucI* and *fucK*. The resulting strain, designated Fuc 2 (Δ*fucI* Δ*fucK* Δ*tpiA*, *fucA* overexpression), was cultivated under the same conditions as those for Fuc 1. Compared with Fuc 1, Fuc 2 exhibited

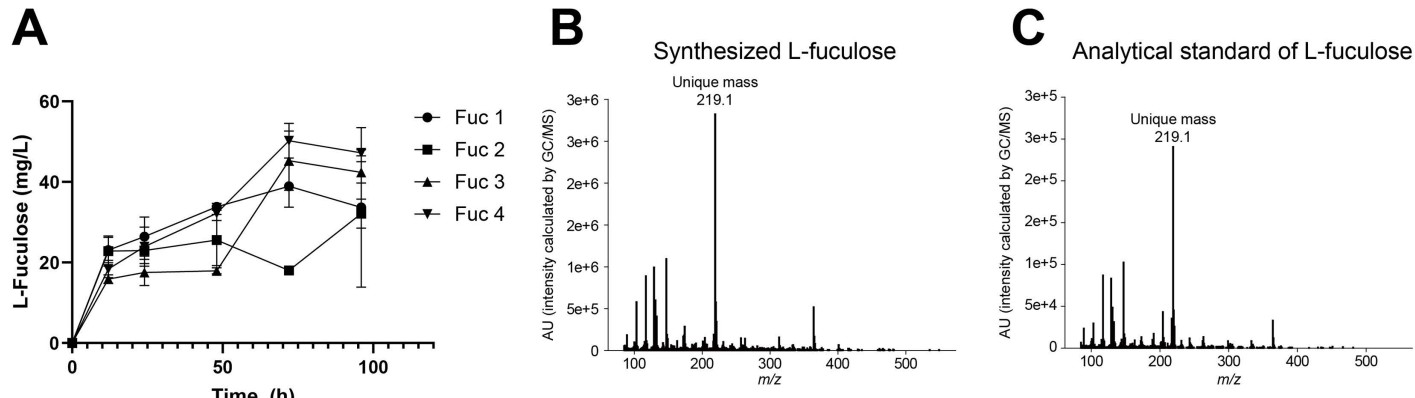

**Fig 3. L-fuculose production during fermentation and confirmation by mass spectrometry. (A)** L-Fuculose production by engineered *E. coli* strains (Fuc 1–4). Data are presented as mean ± standard deviation. **(B)** Mass spectrum of L-fuculose produced by the engineered strain. **(C)** Mass spectrum of the analytical L-fuculose standard.

markedly reduced growth, with glucose being consumed only gradually over 96 h, accompanied by lower biomass formation and acetic acid production (Fig 2B). LAD consumption was also delayed, requiring up to 48 h for complete depletion, whereas conversion into 1,2-PDO occurred steadily throughout the fermentation. L-fuculose production increased gradually during fermentation (Fig 3B), but the final concentration was lower than that obtained with Fuc 1, reaching 32.15 ± 3.62 mg/L, corresponding to a yield of 4.96 ± 0.94%. Deletion of *tpiA* can lead to DHAP accumulation and toxicity, thereby impairing cell growth [16,17]. Consistent with this observation, the severe growth defect observed in Fuc 2 likely resulted from elevated intracellular DHAP levels. Although the deletion of *tpiA* was expected to enhance flux toward L-fuculose, actual L-fuculose production decreased compared to that achieved using Fuc 1, suggesting that DHAP accumulation exerted a toxic effect that outweighed the potential benefits of increased substrate availability.

To further reduce LAD diversion into byproducts, the *fucO* and *aldA* genes were also deleted. The *fucO* encodes a 1,2-PDO oxidoreductase, which converts LAD into 1,2-PDO, whereas *aldA* encodes an aldehyde dehydrogenase, which converts LAD into lactate. The resulting strains were designated as Fuc 3 (Δ*fucI* Δ*fucK* Δ*tpiA* Δ*fucO*, *fucA* overexpression) and Fuc 4 (Δ*fucI* Δ*fucK* Δ*tpiA* Δ*fucO* Δ*aldA*, *fucA* overexpression). Both Fuc 3 and Fuc 4 displayed growth and glucose consumption patterns similar to those of Fuc 2, but LAD depletion was further delayed, with complete consumption occurring only after 72 h (Fig 2). Despite this delay, most LAD was still converted into 1,2-PDO. The L-fuculose production improved substantially compared to that achieved using Fuc 1 and Fuc 2. L-fuculose production increased during fermentation (Fig 3A). Fuc 3 produced 45.30 ± 7.37 mg/L L-fuculose (yield: 6.11 ± 0.99%), whereas Fuc 4 yielded 50.25 ± 4.30 mg/L (yield: 6.78 ± 0.58%). Compared to *Saccharomyces cerevisiae* (used in a previous study), which produced only 1.55 mg/L of L-fuculose [1], the Fuc 4 strain achieved more than a 30-fold improvement, suggesting the potential of *E. coli* as a host for L-fuculose biosynthesis. Although aldA deletion was intended to block the conversion of LAD to lactate, lactate levels were not directly quantified in this study. Therefore, the contribution of aldA deletion to reducing lactate formation remains to be further validated.

These results indicate that elimination of *fucO* and *aldA* reduced LAD catabolism to some extent and prolonged substrate availability, thereby enhancing L-fuculose production. The majority of LAD was still diverted to 1,2-PDO (Fig 2), suggesting that oxidoreductases other than that encoded by *fucO* may contribute to this conversion. In addition, the accumulation of acetate observed during fermentation likely reflects overflow metabolism in *E. coli* under high carbon flux conditions, which can divert carbon away from the desired product and reduce L-fuculose production efficiency [18]. The overall yields remained below 7% of the theoretical maximum, highlighting insufficient FucA activity and limited

dephosphorylation efficiency of sugar phosphatases as critical metabolic bottlenecks. If L-fuculose-1-phosphate is not rapidly dephosphorylated, it can be reconverted into LAD and DHAP by the reverse FucA reaction, further restricting net L-fuculose production. Collectively, these findings suggest that further optimization—such as enhancing FucA and sugar phosphatase activities and identifying additional enzymes involved in LAD-to-1,2-PDO conversion—is required to achieve higher yields of L-fuculose from *E. coli*. Although this study was performed in shake-flask cultures, future studies involving bioreactor-scale fermentation will be important to validate the scalability and industrial relevance of the proposed strategy.

## Conclusions

In this study, a metabolic engineering strategy for L-fuculose biosynthesis in *E. coli* was established by integrating GEM analysis with targeted gene deletions and pathway optimization. GEM analysis determined that the fluxes of LAD, DHAP, and L-fuculose compete, thereby constraining theoretical yields; this result guided the rational design of engineered strains. Sequential deletions of *fucI*, *fucK*, *tpiA*, *fucO*, and *aldA*, together with *fucA* overexpression, enabled direct biosynthesis of L-fuculose from LAD and glucose. The final strain, Fuc 4, produced $50.25 \pm 4.30$ mg/L of L-fuculose, corresponding to 6.78% of the theoretical maximum and representing a > 30-fold improvement over that achieved using our previous microbial platform.

Despite this progress, several bottlenecks remain, e.g., LAD diversion to 1,2-PDO, DHAP toxicity, and limited activities of FucA and sugar phosphatases. Overcoming these challenges through enzyme engineering and elimination of additional competing pathways is critical for further improvement. Overall, this work demonstrates the feasibility of using microbial cell factories for rare sugar biosynthesis and provides a foundation for future process development toward the industrial production of L-fuculose as a functional ingredient for cosmetics, foods, and pharmaceuticals.

## Supporting information

**S1 Table. Vectors and primers used in this study.**
(XLSX)

**S2 Table. Metabolic flux balance analysis for l-fuculose synthesis under aerobic and anaerobic conditions.**
(XLSX)

**S3 Table. Raw dataset of time-course fermentation profiles for fuculose-producing strains. No samples were collected at 6 h for Fuc2–Fuc4. ND values in fuculose indicate loss during derivatization.**
(XLSX)

**S1 Fig. SDS-PAGE analysis of FucA overexpression.** Protein expression was induced with IPTG, and whole-cell lysates were analyzed by SDS-PAGE. A prominent band corresponding to the expected molecular weight of *fucA* was observed in the IPTG-induced sample, whereas this band was absent or significantly weaker in the non-induced control, confirming successful overexpression of *fucA*.
(PNG)

## Acknowledgments

We sincerely appreciate the invaluable assistance of the Green Bio Research Facility Center at the Institutes of Green Bio Science & Technology with the metabolite analysis.

## Author contributions

**Conceptualization:** Gun-Hwi Yeon, Jungyeon Kim.

**Data curation:** Gun-Hwi Yeon, Du-Kyeong Kang, Hyun-Jin Koo, Daewon Go.

**Funding acquisition:** Bong Hyun Sung.

**Investigation:** Gun-Hwi Yeon, Hyun-Jin Koo, Daewon Go.

**Methodology:** Gun-Hwi Yeon.

**Project administration:** Jungyeon Kim, Bong Hyun Sung.

**Resources:** Bong Hyun Sung.

**Software:** Gun-Hwi Yeon.

**Supervision:** Bong Hyun Sung.

**Writing – original draft:** Gun-Hwi Yeon.

**Writing – review & editing:** Gun-Hwi Yeon, Du-Kyeong Kang, Jungyeon Kim, Bong Hyun Sung.

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
