## [Decision Letter · Decision Letter 0]

25 Feb 2026

PONE-D-25-60454 Enhanced production of l-fuculose by Escherichia coli engineered via genome-scale metabolic modeling PLOS One

Dear Dr. Kim,

Thank you for submitting your manuscript to PLOS ONE. After careful consideration, we feel that it has merit but does not fully meet PLOS ONE’s publication criteria as it currently stands. Therefore, we invite you to submit a revised version of the manuscript that addresses the points raised during the review process.   Please submit your revised manuscript by Apr 11 2026 11:59PM. If you will need more time than this to complete your revisions, please reply to this message or contact the journal office at plosone@plos.org. Please include the following items when submitting your revised manuscript:

We look forward to receiving your revised manuscript.

Kind regards,

Jian Xu, Ph.D.

Academic Editor

PLOS One

Journal Requirements:

“This work was supported by National Research Foundation of Korea (NRF) grants funded by the Korean government (MSIT) (RS-2025-23282972, NRF-2022M3J5A1056169, 2021M3A9I5023254, 2019R1A2C1090726, and 2018M3A9H3024746); a National Research Council of Science & Technology grant (No. CAP20024-200) funded by the Korean government (MSIT); the Research Initiative Program of KRIBB; and the Korea Institute of Planning and Evaluation for Technology in Food, Agriculture, and Forestry (IPET) through the Technology Commercialization Support Program funded by the Ministry of Agriculture, Food, and Rural Affairs (MAFRA) (RS-2024-00401586).”

4. Please note that funding information should not appear in any section or other areas of your manuscript. We will only publish funding information present in the Funding Statement section of the online submission form. Please remove any funding-related text from the manuscript.

Additional Editor Comments:

The reviewer has identified several major issues in this study that require substantial revision at this stage. Additionally, the Editor has the following concerns and requests to help the authors further improve their manuscript:

1) Please provide growth curves and microscopic images (e.g., confocal microscopy) for all generated strains alongside the appropriate control samples.

2) How were the knockout (KO) and knock-in (KI) strains verified? Any PCR validation data must be included, either as a main or supplemental figure. Furthermore, evidence supporting the overexpression of fucA should also be provided.

3) The discussion section is currently unsatisfactory and requires more careful revision. Overstatements should be strictly avoided and conclusions must be supported by the results presented in this study.

Reviewer's Responses to Questions

**Comments to the Author**

1. Is the manuscript technically sound, and do the data support the conclusions?

Reviewer #1: Yes

2. Has the statistical analysis been performed appropriately and rigorously? 

Reviewer #1: Yes

3. Have the authors made all data underlying the findings in their manuscript fully available?

Reviewer #1: Yes

4. Is the manuscript presented in an intelligible fashion and written in standard English?

Reviewer #1: Yes

5. Review Comments to the Author

Reviewer #1: Authros focus on constructing an L-fuculose producing strain by deleting fucI, fucK, tpiA, fucO, and aldA and overexpressing fucA. The final engineered strain produced 50.25 mg/L of l-fuculose via 120-h cultivation. Some questions and suggestions are listed as follows.

(1) It is suggested to show the dynamic production of l-fuculose during the fermentation.

(2) Considerable acetate was accumulated. Discussion is required.

(3) AldA was deleted to block conversion of LAD to lactate. Was accumulation of lactate detected before and after aldA deletion?

(4) fucA::T7, what’s the meaning of T7 here? Which promoter was applied to express fucA?

(5) It is suggested to perform the fermentation in a bioreactor to further indicate the significance of this study.

6. PLOS authors have the option to publish the peer review history of their article (what does this mean?). If published, this will include your full peer review and any attached files.

Reviewer #1: No

---

## [Author Response · Author response to Decision Letter 1]

22 Apr 2026

Response to Reviewers’ Comments

Dear Editor and Reviewer,

We sincerely thank the editor and reviewer for their careful evaluation and constructive comments. We have revised the manuscript accordingly and believe that these changes have significantly improved the clarity and quality of our work. Below, we provide a point-by-point response to all comments.

<Editorial Comment>

[Comments]

Comment 1. Please provide growth curves and microscopic images (e.g., confocal microscopy) for all generated strains alongside the appropriate control samples.

Response: We thank the editor for this valuable suggestion. Growth curves for all engineered strains have been included in the revised manuscript (Fig. 2 and 3).

Regarding microscopic analysis, while such data may provide additional insights into cellular morphology, the genetic modifications introduced in this study were not expected to induce significant morphological changes. As the primary focus of this study was on metabolic flux redistribution and L-fuculose production, and growth characteristics were already evaluated through growth curves, we believe that the current dataset sufficiently supports our conclusions. Microscopic analysis may be considered in future studies.

[Revision: Figs. 2 and 3]

Comment 2. How were the knockout (KO) and knock-in (KI) strains verified? Any PCR validation data must be included. Furthermore, evidence supporting the overexpression of fucA should also be provided.

Response: We thank the editor for this important comment. The knockout strains were verified by PCR during strain construction, as described in the Methods section (Construction of mutants). In addition, evidence supporting the overexpression of fucA has been added to the revised manuscript. As shown in Fig. S1, SDS-PAGE analysis revealed a prominent band at the expected molecular weight of FucA following IPTG induction, confirming successful overexpression.

[Revision: Fig. S1]

Comment 3. The discussion section is currently unsatisfactory and requires more careful revision. Overstatements should be strictly avoided and conclusions must be supported by the results presented in this study.

Response: We thank the editor for this valuable comment. The Discussion section has been carefully revised to avoid overstatements and to ensure that all conclusions are strictly supported by the experimental results. The language has been moderated throughout to better reflect the scope and limitations of the study.

[Revision: Lines 249-254]

Compared to Saccharomyces cerevisiae (used in a previous study), which produced only 1.55 mg/L of L-fuculose [1], the Fuc 4 strain achieved more than a 30-fold improvement, suggesting the potential of E. coli as a host for L-fuculose biosynthesis. Although aldA deletion was intended to block the conversion of LAD to lactate, lactate levels were not directly quantified in this study. Therefore, the contribution of aldA deletion to reducing lactate formation remains to be further validated.

<Reviewer 1>

[Comments]

Comment 1. It is suggested to show the dynamic production of L-fuculose during the fermentation.

Response: We thank the reviewer for this helpful suggestion. The time-dependent production of L-fuculose during fermentation has now been incorporated into the revised manuscript and is presented in Fig. 3. Corresponding descriptions have been added in the Results section.

[Revision: Lines 203-207; Fig. 3]

Fig 3. L-fuculose production during fermentation and confirmation by mass spectrometry. (A) L-Fuculose production by engineered E. coli strains (Fuc 1–4). Data are presented as mean ± standard deviation. (B) Mass spectrum of L-fuculose produced by the engineered strain. (C) Mass spectrum of the analytical L-fuculose standard.

LAD was rapidly depleted within the first 12 h, whereas 1,2-PDO began to accumulate from 12 h, with its concentration increasing gradually until the end of fermentation. L-fuculose production was detected after 12 h and gradually increased throughout fermentation, reaching 38.92 ± 5.19 mg/L (Fig 3A). GC/MS analysis of the culture supernatant confirmed the presence of L-fuculose (Fig 3B, C).

Comment 2. Considerable acetate was accumulated. Discussion is required.

Response: We thank the reviewer for this important comment. A discussion on acetate accumulation has been added to the revised manuscript. We now describe that acetate accumulation likely reflects overflow metabolism in E. coli under high carbon flux conditions and may reduce carbon availability for L-fuculose biosynthesis.

[Revision: Lines 258-261]

In addition, the accumulation of acetate observed during fermentation likely reflects overflow metabolism in E. coli under high carbon flux conditions, which can divert carbon away from the desired product and reduce L-fuculose production efficiency [18].

Comment 3. AldA was deleted to block conversion of LAD to lactate. Was accumulation of lactate detected before and after aldA deletion?

Response: We thank the reviewer for this insightful comment. Although aldA deletion was intended to block the conversion of LAD to lactate, lactate levels were not directly quantified in this study. This limitation has been acknowledged in the revised manuscript, and future studies will be required to further evaluate the contribution of aldA deletion to lactate reduction.

[Revision: Lines 251-254]

Although aldA deletion was intended to block the conversion of LAD to lactate, lactate levels were not directly quantified in this study. Therefore, the contribution of aldA deletion to reducing lactate formation remains to be further validated.

Comment 4. fucA::T7, what’s the meaning of T7 here? Which promoter was applied to express fucA?

Response: We thank the reviewer for this comment. The description has been clarified in the revised manuscript to indicate that fucA was expressed under the control of the T7 promoter.

[Revision: Lines 196-200]

In addition, fucA was overexpressed under the control of the T7 promoter to catalyze the aldol condensation of LAD and DHAP to L-fuculose-1-phosphate. Overexpression of fucA was confirmed by SDS-PAGE analysis following IPTG induction (Fig. S1), showing a distinct band at the expected molecular weight. The resulting strain, designated Fuc 1 (ΔfucI ΔfucK, fucA overexpression), was cultivated in glucose–LAD medium for 96 h.

Comment 5. It is suggested to perform the fermentation in a bioreactor to further indicate the significance of this study.

Response: We thank the reviewer for this valuable suggestion. While the present study was conducted in shake-flask cultures, we have added a statement in the Discussion section indicating that future studies involving bioreactor-scale fermentation will be important to evaluate scalability and industrial applicability.

[Revision: Lines 268-271]

Although this study was performed in shake-flask cultures, future studies involving bioreactor-scale fermentation will be important to validate the scalability and industrial relevance of the proposed strategy.

---

## [Decision Letter · Decision Letter 1]

29 Apr 2026

Enhanced production of l-fuculose by Escherichia coli engineered via genome-scale metabolic modeling

PONE-D-25-60454R1

Dear Dr. Kim,

We’re pleased to inform you that your manuscript has been judged scientifically suitable for publication and will be formally accepted for publication once it meets all outstanding technical requirements.

Kind regards,

Jian Xu, Ph.D.

Academic Editor

PLOS One

Additional Editor Comments (optional):

Reviewers' comments:

Reviewer's Responses to Questions

**Comments to the Author**

1. If the authors have adequately addressed your comments raised in a previous round of review and you feel that this manuscript is now acceptable for publication, you may indicate that here to bypass the “Comments to the Author” section, enter your conflict of interest statement in the “Confidential to Editor” section, and submit your "Accept" recommendation.

Reviewer #1: All comments have been addressed

2. Is the manuscript technically sound, and do the data support the conclusions?

Reviewer #1: Yes

3. Has the statistical analysis been performed appropriately and rigorously? 

Reviewer #1: Yes

4. Have the authors made all data underlying the findings in their manuscript fully available?

Reviewer #1: Yes

5. Is the manuscript presented in an intelligible fashion and written in standard English?

Reviewer #1: Yes

6. Review Comments to the Author

Reviewer #1: The manuscript has been well revised according to the comments. I recommend that it should be accepted.

7. PLOS authors have the option to publish the peer review history of their article (what does this mean?). If published, this will include your full peer review and any attached files.

Reviewer #1: No

---

## [Editor Report · Acceptance letter]

PONE-D-25-60454R1

PLOS One

Dear Dr. Kim,

I'm pleased to inform you that your manuscript has been deemed suitable for publication in PLOS One. Congratulations! Your manuscript is now being handed over to our production team.

Kind regards,

on behalf of

Dr. Jian Xu

Academic Editor

PLOS One